# Functional Diversification and Structural Origins of Plant Natural Product Methyltransferases

**DOI:** 10.3390/molecules28010043

**Published:** 2022-12-21

**Authors:** Audrey Lashley, Ryan Miller, Stephanie Provenzano, Sara-Alexis Jarecki, Paul Erba, Vonny Salim

**Affiliations:** 1Department of Biological Sciences, Louisiana State University, Shreveport, LA 71115, USA; 2School of Medicine, Louisiana State University Health New Orleans, New Orleans, LA 70112, USA; 3School of Medicine, Louisiana State University Health Shreveport, Shreveport, LA 71103, USA

**Keywords:** methyltransferase, plant natural products, bioactive molecules, enzyme, structure, pharmaceuticals

## Abstract

In plants, methylation is a common step in specialized metabolic pathways, leading to a vast diversity of natural products. The methylation of these small molecules is catalyzed by *S*-adenosyl-l-methionine (SAM)-dependent methyltransferases, which are categorized based on the methyl-accepting atom (*O*, *N*, *C*, *S,* or *Se*). These methyltransferases are responsible for the transformation of metabolites involved in plant defense response, pigments, and cell signaling. Plant natural product methyltransferases are part of the Class I methyltransferase-superfamily containing the canonical Rossmann fold. Recent advances in genomics have accelerated the functional characterization of plant natural product methyltransferases, allowing for the determination of substrate specificities and regioselectivity and further realizing the potential for enzyme engineering. This review compiles known biochemically characterized plant natural product methyltransferases that have contributed to our knowledge in the diversification of small molecules mediated by methylation steps.

## 1. Introduction

Methylations, a process universal within all organisms, are responsible for extensive cellular modulations. These reactions are typically catalyzed by methyltransferases that require cofactor *S*-adenosyl-l-methionine (SAM), resulting in the formation of the methylated product and *S*-adenosyl-l-homocysteine (SAH) [1]. More commonly known for their roles in epigenetics, methyltransferases are also involved in natural product biosynthesis in plants. Plant natural product methyltransferases (PNPMTs) contribute to the diversification of specialized metabolites with functions such as pigments [2,3], antioxidants [4], signal transducer, and plant defense response [5,6,7,8]. Phenolic compounds, specifically, support plant growth and provide additional advantages to tolerate environmental stresses against insects and microbial invasion [9,10].

PNPMTs and their subsequent methylated products also display pharmaceutical properties for humans. Compared to their non-methylated counterparts, methylated plant natural products exhibit different chemical properties, and thereby alter biological activity. For instance, methylated resveratrol was shown to decrease its genotoxicity level compared to non-methylated resveratrol, which has been reported to cause chromosome aberration [11]. Furthermore, methylated resveratrol displays improved bioavailability and overall bioactivity [12,13]. Methylation also influences the binding of small molecules to a human neurotransmitter receptor. For example, as an intermediate in the morphine pathway, the *O*-methylated benzylisoquinoline (BIA) thebaine exhibits stimulatory effects and weaker analgesic properties compared to morphine, which lacks *O*-methylated groups [14]. Similarly, the methylated monoterpene indole alkaloid (MIA) ibogaine is less polar than its non-methylated counterpart, noribogaine, and is far more readily sequestered to the lipophilic compartment of the brain [15,16].

In recent years, advanced -omics have enabled the discoveries of new plant natural product methyltransferases. A number of PNPMTs have been elucidated by X-ray crystallography, illuminating the SAM and substrate-binding domains. Canonically, based on structures, all natural-product methyltransferases across kingdoms belong to Classes I and III [1]. PNPMTs, bearing the Rossmann fold-binding site for SAM, are categorized as Class I, while Class III is typically found in bacteria and associated with membranes [1,17,18,19]. The remaining classes of methyltransferases have been structurally characterized by the domains that participate in macromolecular (DNA, RNA, and protein) methylation for epigenetic regulation and have been reviewed elsewhere [19,20,21]. In the past twenty years, the emergence of synthetic biology and biotechnological advancements in the heterologous production of plant enzymes in microbial systems have accelerated the gene identification and functional characterization of plant natural product enzymes, including methyltransferases. Here, we highlight how the biochemical characterizations of PNPMT genes integrated with the structural knowledge could lay the foundations for more extensive enzyme engineering efforts. In this review, we focus on Class I, SAM-dependent plant natural product methyltransferases (PNPMTs), summarizing their structure, function, and application for metabolic engineering to alter their enzyme specificity and diversification of plant specialized metabolites.

## 2. Classification of PNPMTs Based on Methyl Acceptor

Methyltransferases are categorized based on the substrate atoms *O*-, *N*-, *C*-, *S*-, and *Se*- that accept the methyl group. Though rare, a few reported PNMPTs were noted for their ability to transfer methyl groups to more than one acceptor, such as those methylating halides and thiocyanates [22]. Among all plant natural product methyltransferases, *O*-directed methyltransferases (OMTs) are the most abundant to date [1]. OMTs target the hydroxyl groups of small molecules, such as alkaloids, lignin precursors, simple phenols, phytoalexins, phytohormones, and flavonoids, to produce their methylated form. This also includes the hydroxyl moiety of the carboxyl group, which can be found in salicylic acid [5,23], jasmonic acid [7] and iridoid loganic acid [24]. OMTs are also further characterized into two subclasses based on their dependence on or independence of a divalent cation. The cation-dependent OMTs, also known as caffeoyl-CoA OMTs (CCoAOMTs), typically have a lower subunit molecular weight of 26–30 kDa. CCoAOMTs are mainly involved in lignin, phytohormone, and scent metabolism [16,25]. Comparatively, the cation-independent OMTs, also known as caffeic acid OMT (COMT), range between 37 and 43 kDa and are primarily active in phenylpropanoid and alkaloid biosynthesis [26].

The abundance of OMTs is also seen through the incredible diversity of their methoxylated products (Figure 1). For instance, plant alkaloid OMTs have been well-characterized, including the monoterpene indole alkaloid in *Catharanthus roseus* to synthesize vindoline, which is then dimerized with catharanthine to form anticancer vinblastine [27] and monoterpene isoquinoline alkaloid in *Psychotria ipecacuanha* [28]. Notably, out of all known alkaloid OMTs to date, the majority of them are involved in benzylisoquinoline alkaloid biosynthesis (Table 1). Further characterization of plant alkaloid OMTs, known as norbelladine 4′OMTs, has been reported in Amaryllidaceae alkaloid biosynthesis from *Narcissus* spp and *Lycoris aurea* [29,30]. Although heterocyclic nitrogenous methoxypyrazines are not characterized as alkaloids, the OMTs involved in their pathway have been reported [31,32,33].

The majority of plant natural product OMTs have been characterized as part of phenylpropanoid-related biosynthetic pathways, which include flavonoids. Various types of flavonoids (flavonols, flavones, isoflavones, flavanols, and anthocyanins) play significant physiological roles in plants, typically in response to biotic and abiotic stresses [136]. For instance, the methoxylated flavonoid tricin in rice serves as a plant defense response against insect herbivores [137]. While flavonoid-OMTs generally do not require a cation for catalyzation, cation-dependent flavonoid-OMTs have been reported. Specific cation-dependent flavonoid-OMTs include the anthocyanin-OMT and an (iso) flavonol-6-OMT, which are multifunctional in methylating caffeoyl-CoA and associated flavonoids [70,90,95,138] (Table 1).

The position and number of methyl groups determine the structural diversity of flavonoids, though these are restricted by the placement of hydroxyl groups [136]. *O*-methylation can occur at any position theoretically, however, 7- and 4′-methoxylation are most prevalent among flavonoids; interestingly, a number of polymethoxylated flavonoids have also been reported [136]. Although the majority of flavonoid-OMTs are active towards aglycone substrates, anthocyanin 3′OMT and 5′OMT were later characterized and displayed strong preference for glycosylated substrates [138].

While OMTs are generally regiospecific, plant OMTs involved in phenylpropanoid and flavonoid biosynthesis are less selective and catalyze sequential methylations of almost identical substrates [1,65,107,139]. A small group of plant OMTs has demonstrated strong enzymatic specificity for a single structure, but overall, plant OMTs tend to accept a wide span of substrates, with varying substrate preferences, including flavonoids, lignin precursors, and alkaloids [48,140]. For example, an alkaloid OMT, 10-hydroxycamptothecin OMT in *Camptotheca acuminata*, was strongly preferred to methylate 10-hydroxycamptothecin but was also capable of methylating flavonoids, stilbenes, and caffeic acids [48].

Plant natural product OMTs have also been cloned and biochemically characterized from several BIA-producing species. In the most studied BIA-producing organism, *Papaver somniferum*, once (*S*)-norcoclaurine is formed by the condensation of dopamine and 4-hydroxyphenylacetaldehyde, the hydroxyl group in the isoquinoline moiety is methylated to produce (*S*)-coclaurine. Further characterization of similar cDNAs encoding this methyltransferase has been reported in six species [34,141,142,143,144]. Moreover, the crystal structure of norcoclaurine-6-*O*-methyltransferase from *Thalictrum flavum* (Tf6OMT) has been reported (PDB accession number: 5ICC) [143]. Additional OMTs in the BIA pathway contribute to the final step in the central pathway to produce (*S*)-reticuline by methylating 3′-hydroxy-*N*-methylcoclaurine 4′-*O*-methyltransferase (4′OMT). The corresponding transcripts encoding 4′OMT-like enzymes have also been reported in three species [34,141,142,144,145,146]. A number of these BIA OMTs are remarkably substrate-specific, such as norreticuline 7-*O*-methyltransferase, which introduces a third methoxy group to produce (*S*)-norlaudanine in papaverine biosynthesis [38,141]. Multiple *O*-methylation is more common in BIA biosynthetic pathways, and the additional transcripts of similar enzymes have also been reported, including reticuline 7-*O*-methyltransferase and scoulerine 9-*O*-methyltransferase [39,141,142,144,147,148,149]. In other cases, a similar enzyme to biochemically characterized scoulerine 9-*O*-methyltransferase preferentially methylates the 2-hydroxyl of quaternary protoberberine and columbamine in only *Coptis japonica* [150].

Though not as abundant as the OMTs, *N*-directed methyltransferases (NMTs) still offer a respectable diversity of natural products. The *N*-methylated phytochemicals consist of primary amines, secondary amines found in indoles and imidazoles, tertiary amines, or more complex alkaloids [1] (Figure 2). Most *N*-directed methyltransferases have been cloned from alkaloid-producing plants. In particular, the caffeine biosynthetic pathway in xanthosine involves a series of *N*-methylation steps that serve as a model to study alkaloid biosynthesis [151,152]. The biosynthetic genes involved were cloned from *Camellia sinensis* [152,153], *Coffea arabica* [54,55,57,58,153] *Paullinia cupana* [59], *Theobroma cacao* [154], and *Citrus sinensis* [151]. Remarkably, caffeine and its xanthine precursors have evolved independently in a number of flowering plants using either of the two biochemical pathways through caffeine synthase or xanthine methyltransferase-like enzymes, as exhibited in chocolate, citrus, and guarana plants [151]. Based on coding sequences, xanthine NMTs were determined to belong to the SABATH (salicylic acid, benzoic acid, theobromine synthase) family of methyltransferases, which exclusively utilize the “proximity and desolvation effects” [1,5], rather than general acid-base catalysis or cation-dependent mechanisms. Since SABATH methyltransferases generally methylate oxygen atoms, it is likely that a recent change in the function of xanthine alkaloid-producing species occurred [16]. A recent study hypothesized that despite stemming from different lineages, xanthosine methyltransferases have convergent evolutionary histories [16,151].

A number of NMTs involved in monoterpene indole alkaloid biosynthetic pathways have been characterized, including those involved in vindoline production in *Catharanthus roseus* [49], *N*-methylation of picrinine from the Apocynaceae family [51], and ajmaline synthesis [52]. NMTs have contributed significantly to the diversification of benzylisoquinoline alkaloids (BIA), and several NMTs have been characterized in nine *Ranunculales* species [16].

*C*-directed methyltransferases (CMTs) have been well characterized and primarily participate in specialized metabolism by methylating aliphatics, such as tocopherol [116,117,118] and sterol [120,121,122,123]. Notable examples highlighted in this review are illustrated in Figure 2.

*S*-directed methyltransferases (SMTs) are generally involved in the production of volatile halogen and sulfur compounds as well as thiocyanate [1]. A few examples of characterized SMTs in plants have been reported, including those from *C. roseus* [132]. A thiocyanate methyltransferase found in *Arabidopsis thaliana* that synthesizes methylthiocyanate [124] was later structurally analyzed [22]. Halide/bisulfide methyltransferase activity was also detected in the leaves of *Brassica oleracea* [155]. Some MTs exhibit high specificity for halides or bisulfides, known as halide ion methyltransferases (HMTs) and halide/thiol methyltransferases (HTMTs). HMTs and HTMTs are found in coastal trees, grasses, and several agricultural plants, with particularly high levels of activity in *Raphanus sativus* (daikon radish), *Oryza sativa* (paddy rice) *Triticum aestivum* (wheat), and *Cyathea lepifera* (fern) [125]. These HMT and HTMT genes were later cloned [156,157]. HMTs have also been biochemically characterized in broccoli with high substrate specificity toward homocysteine [158]. Lyi and colleagues also reported a selenocysteine methyltransferase that produces *Se*-methylselenocysteine [135]. Interestingly, these two genes play roles in sulfur and selenium metabolism in broccoli. Another SAM-dependent HTMT was cloned from *Raphanus*, and upon biochemical testing, exhibited high specificity for iodide, bisulfide, and thiocyanate [125]. Plant SMTs have also been characterized in the production of brassinin in cruciferous vegetables [131,159]. A comprehensive list of biochemically characterized PNPMTs with some structurally characterized features is detailed in Table 1.

## 3. The Identification of Plant Natural Product Methyltransferase (PNPMT) Genes

All known Class I PNPMTs consist of an alternating α-helix/ß-strand structure with associated monomeric molecular masses of 25 to 55 kDa [1]. Bioinformatic analyses to identify putative PNPMT genes are primarily based on a series of conserved motifs, with heavy reliance on the core Rossmann fold [1,25]. Within the Rossmann fold, a nine-residue amino acid incorporating a glycine-rich structure “GxGxG” signature sequence corresponding to a SAM-binding motif is found in SAM-dependent MTs (red boxes in Figure 3). Although these consensus sequences (V/I/L)(L/V)(D/E)(V/I)G(G/C)G(T/P)G [160] contain substitutions, the amino acid sequences are typically better aligned within the same subclasses (cation-dependent, cation-independent, SABATH, BIA).

Amino acid sequences of PNPMTs from separate subclasses of cation-dependent, cation-independent, SABATH, and BIA MTs were chosen from the list of biochemically characterized plant natural product MTs (Table 1) to construct a phylogenetic tree (Figure 4). The genes within the same subclasses share high sequence similarity, while several of these highly homologous MTs have been reported to be involved in the biosynthesis of similar classes of metabolites, including clusters of BIA OMTs/NMTs and alkaloid biosynthetic genes (Ca10OMT, IpeOMT1, Vm16OMT, Cr16OMT) [16,48]. Sequence analysis and the phylogenetic results confirmed the functional classification of cation-dependent, cation-independent, and SABATH families. Class I cation-dependent bacteria OMT is included to compare with plant cation-dependent OMTs (Figure 4).

## 4. Structural Biochemistry of PNPMTs

Protein X-ray crystallography has allowed for the extensive structural elucidation of numerous Class I methyltransferases. The resulting X-ray crystal structures have offered great insight into the general structures of these methyltransferases and have determined that they remain highly consistent, despite the primary amino acid sequences having low similarity [162]. Nevertheless, the primary amino acid sequences are still necessary to conduct phylogenetic analysis as they yield clarity into the evolutionary relationships of all PNPMTs. Furthermore, increased genomic data and the availability of coding sequences facilitate the molecular characterization or determinants of function [16,151]. For example, characterized active sites that accept xanthine have distinguishing features that promote attachment with xanthine molecules [152,154]. In particular, the hydrophobic pocket and specific residues affecting substrate specificity, encourage such xanthine attachment, later confirmed by site-directed mutagenesis [16,151,154,163]. In plant OMTs, structural analysis often involves dimers. Most OMTs form homodimers [92], though a BIA OMT was reported to form a heterodimer [164]. Typically, a dimerization interface is present as several helices actively participate in substrate binding. This dimerization interface also constitutes a hydrophobic “back wall” [92], though its participation in substrate binding varies among plant OMTs [16]. The analysis of primary amino acid sequences revealed that the *N*-terminus domain contributes to the dimerization with partial involvement in substrate-binding within the hydrophobic pocket, while the *C*-terminus domain is involved in SAM-binding via the Rossmann fold.

Class I PNPMTs are characterized by a conserved domain known as the Rossmann fold. The Rossmann fold is defined by a seven-stranded ß-sheet, commonly ordered 3 2 1 4 5 7 6, with the seventh strand in an antiparallel position to the remaining strands and three α-helices on each side of the ß-sheet [17,165].

Recognized as the defining features of all SAM-dependent methyltransferases, six sequentially ordered motifs (I-VI) either contribute to the formation of the SAM-binding pocket or have interactions with SAM itself. [1,18,25,166] (Figure 5). Motif I consists of ß-strand I and the adjoining loop, and located at the *N*-terminus is a glycine-rich sequence (GxGxG) with one or two glycines interacting with SAM [18]. Generally, this glycine-rich sequence remains conserved throughout Class I, but there is a minority of cases where alanine residues replaced the glycines [18]. Motif II consists of ß-strand II and the following helix, and there is either an aspartate or glutamate residue located at the *C*-terminus [18,166]. The helix is structurally conserved and directly contributes to the SAM-binding pocket [18]. Motif III is involved in SAM binding. This motif possesses a hydrophilic amino acid at the *N*-terminus of ß-strand III, and a partially conserved glycine residue at the *C*-terminus [18]. Motif IV is composed of ß-strand IV and the adjoining loops. Located at the *C*-terminus is a partially conserved acidic residue [1]. Motif V consists of a helix following strand IV, although this motif does not typically interact with SAM [18]. However, in a few cases, this motif may act as a site for hydrophobic side chains, securing SAM’s adenine moiety [166]. Finally, motif VI, very uncommonly interacting with SAM, is characterized by strand V and its preceding loop, as well as a conserved glycine residue, followed by two hydrophobic residues at the beginning of the strand [18]. Interestingly, among all subclasses, the residues of the motifs were highly conserved, although SAM’s position with its binding pocket varied depending on the subclass [18,25,166] (Figure 5).

The Rossmann fold is a multifunctional domain of the PNPMT and other Class I methyltransferases alike—not only is the Rossmann fold able to house the methyl donor SAM, but it also has influences on the substrate-binding pocket, and thus, the orientation of the methyl acceptor atom [5]. While the Rossmann-fold domain is widely conserved across all Class I methyltransferases, structural tweaks allow for some degree of promiscuity. In contrast to DNA methyltransferases, which are designed to identify the general features of macromolecular substrates due to their widely conserved active sites [19], PNPMTs generally exhibit higher levels of specificity [1]. A number of important amino acid residues that contribute to this specificity have been identified via structural-based analysis and confirmed by site-directed mutagenesis in several cases, including in caffeoyl-CoA methyltransferases [103], chalcone and isoflavone methyltransferase [92], and benzylisoquinoline **[16,167]**.

In some alkaloid biosynthesis studies, heterodimers have been suggested to participate in the catalysis of new substrates. From *Thalictrum tuberosum*, the heterodimers of four OMTs accepted substrates, such as catechols, hydroxycinnamates, and alkaloids, exhibit far more promiscuity than their homodimer counterparts [47]. A missing step in noscapine biosynthesis in *Papaver somniferum* was filled with a then newly discovered heterodimer consisting of PsSOMT2 and PsSOMT/Ps6OMT [164]. Due to the degree of promiscuity within alkaloid heterodimers, understanding their functionalization has the potential to uncover new biochemical processes [136]. Overall, the advances in the structural biochemistry of PNMPTs have highlighted the degree of specificity that both conserves the SAM-binding pocket and provides structural determinants of functions and their involvements in plant natural product biosynthesis.

## 5. Engineering Plant Natural Product Methyltransferases (PNPMTs)

Methyltransferases are attractive enzymes to manipulate for the purpose of altering metabolic pathways. The regioselectivity of PNPMTs is relatively more favorable in comparison to organic synthesis in which protecting groups are often added to produce specific methylated products with a further disadvantage of low yield [1]. The structure–function analysis has been extensively applied for enzyme engineering, which involves mutagenesis approaches to substitute their amino acids to obtain the desired methylated products. For example, a poplar flavonoid-OMT mutant with amino acid Asp257Gly was reported to methylate the 3-hydroxyl group compared to its native selectivity in methylating the 7-hydroxyl group [168]. In another case, variants of *Vitis vinifera* resveratrol OMT with point mutations were generated and resulted in modified native substrate specificity to increase the yield of pinostilbene (monomethylated resveratrol), rather than to produce dimethylated resveratrol [169]. In addition to shifting in the regiospecificity of methylation and/or modified substrate specificity, the promiscuity of PNPMTs has been proven to be more versatile, as the biosynthetic pathway can be altered with a possibility of yielding an unknown molecule via de novo pathways, and thus, increasing the diversity of plant natural products [1,170]. In most cases, beneficial mutations are identified by *in-silico* methods combined with a high-throughput screening assay to accelerate the engineering process. Zhao and colleagues rationally designed *Peucedanum praeruptorum* bergaptol *O*-methyltransferase to increase the production of bergapten with improved pharmacological properties, especially for increasing pigmentation levels [171]. One Val320Ile mutant protein was particularly reported to be promising for further applications in treating depigmentation disorder [171]. In the absence of crystal structures, computer-guided analysis to analyze the active site and the catalytic mechanisms of plant natural product methyltransferases can provide alternative manufacturing strategies via metabolic engineering approaches in microbial systems.

A number of de novo biosynthesis pathways that produce methylated natural products have been successfully designed in microbial systems. For instance, dimethylated resveratrol (pterostilbene) was produced in an engineered *Escherichia coli* strain expressing *Arabidopsis* caffeic acid/resveratrol *O*-methyltransferase. The intracellular pool of l-tyrosine, a precursor molecule in the stilbene biosynthetic pathway, was improved in this engineered strain grown in l-methionine-rich media to increase the SAM supply [172]. This approach overcomes a central issue in engineering methyltransferases, which is cofactor regeneration, as SAH is a known inhibitor for methyltransferase activities and its accumulation can be toxic for the microbial hosts [1,170]. Although the incorporation of native SAM recycling pathways involving multiple enzymatic reactions can be challenging to integrate within the host systems, deregulation of the SAM pathway was shown to successfully increase the production of *O*-methylated products. *MetJ* or Met repressor in bacteria is often a target for silencing or deletion in order to regulate methionine production. Silencing of the *MetJ* gene via CRISPRi enhances the production of *O*-methylated anthocyanin in *Escherichia coli* [173]. Similarly, deleting *MetJ* was also effective in shuttling the pool of SAM in an engineered *Escherichia coli* strain that overexpressed methionine biosynthetic pathway genes, especially in improving vanillate production [174].

The engineering of methyltransferases has advanced beyond its ability in transferring a methyl group. The development of SAM analogs has allowed for the diversification of small molecules. In particular, a mutated halide methyltransferase from *Arabidopsis thaliana* efficiently transferred ethyl-, propyl-, and allyl-moieties to SAH in order to produce SAM analogs with the respective alkyl groups [175]. Recently, efforts in developing SAM analogs to diversify active biomolecules have been accelerated after the discovery of the naturally occurring carboxy-*S*-adenosylmethionine pathway involved in tRNA modification [176]. Interestingly, coclaurine *N*-methyltransferase from *Coptis japonica* was selected to test the possibility of plant methyltransferases catalyzing the reaction using an alternative cofactor of carboxy-SAM [177]. By comparing its SAM-binding residues with the crystal structure tRNA carboxymethyltransferase, which highly prefers carboxyl-SAM, coclaurine *N*-methyltranferase was successfully engineered to accept carboxy-SAM over SAM to produce carboxymethylated tetrahydroisoquinoline [177]. While the use of SAM analogs in the diversification of plant small molecules seems likely to be effective at smaller scales, more innovations are still required to increase efforts in their regeneration and efficient synthesis.

Although microbial systems remain excellent hosts for engineering plant methyltransferases, plant hosts are still relevant to consider, especially in obtaining natural products for industrial and specific agricultural applications. Engineered monolignol 4-*O*-methyltransferase, for example, was useful for reducing lignin content for the effective production of liquid biofuels, a desired utilization of cellulosic fiber, while it allows for the diversification of possible active compounds in transgenic plants [178]. Engineered plant host systems also provide additional platforms for extending the natural product biosynthetic pathways beyond methylated metabolites, as novel compounds with unique biological activities might be accumulated. Overall, the versatility of plant natural product methyltransferases can inspire future efforts in enzyme and metabolic engineering for improved capabilities in widening the diversity of bioactive compounds with pharmacological properties.

## 6. Conclusions and Future Directions

The structural elucidation of PNPMTs has heavily augmented our knowledge of how SAM-dependent methyltransferases contribute to the diversification of small molecules in the plant kingdom. The increasing number of structure determinations, along with the elucidation of their architecture, has significantly provided insights into their molecular evolution. A comprehensive representation of biochemically characterized PNPMTs demonstrates not only the structural diversity of Class I of SAM-dependent methyltransferases but also provides a point of reference to search for other PNPMTs that have yet to be discovered. With the continuous advances in next-generation sequencing and progress in -omics-based technologies, we anticipate more PNPMT genes will be discovered. While only less than 20% of the PNPMTs that we compiled have been structurally elucidated, we expect more reports on the protein crystal structures of PNPMTs. This knowledge of structural biochemistry will benefit the development of machine learning algorithms for more advanced prediction of protein folding, gene annotation, and functional validation. By delineating the substrate specificity and structural attributes of PNPMTs, we have provided an overview of the recently expanded strategies to facilitate the development of enzyme and metabolic engineering. We also anticipate that all these advancements will give rise to further diversification and increased production of methylated natural products with enhanced pharmacological properties as human therapeutics. Ultimately, the complete knowledge of molecular and structural determinants of PNPMT function will reveal important aspects of the catalytic mechanism and inform metabolic engineering efforts, thus generating novel biocatalysts for wide applications in biotechnology.

## Figures and Tables

**Figure 1 molecules-28-00043-f001:**
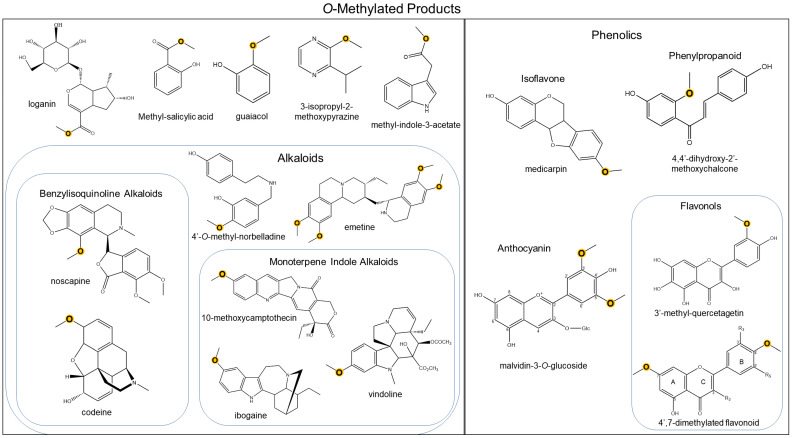
Representation of chemical diversity generated by *O*-directed methyltransferases. Methylated atoms are highlighted in yellow.

**Figure 2 molecules-28-00043-f002:**
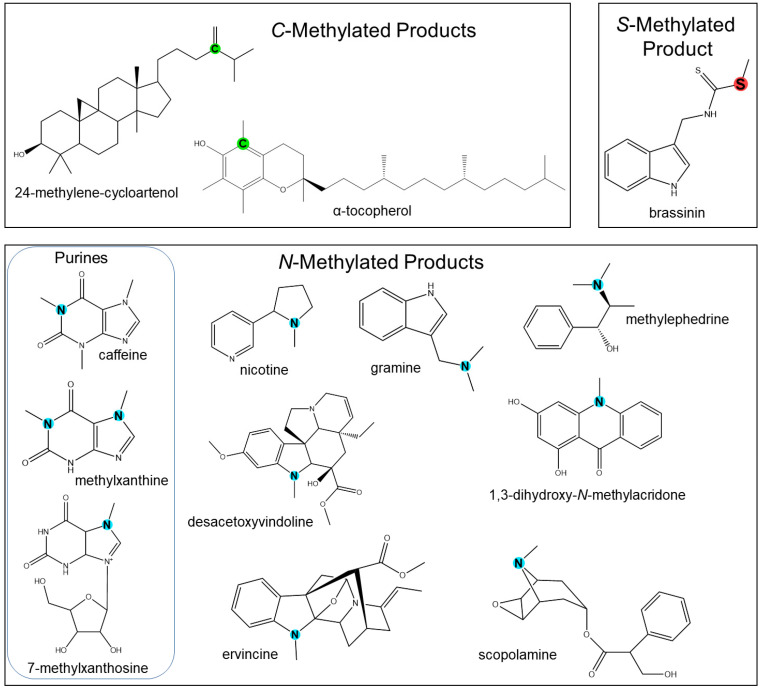
Representation of chemical diversity generated by *N*-, *C*-, *S*-directed methyltransferases. Methylated atoms are highlighted in blue (*N*-), green (*C*-), and red (*S*-).

**Figure 3 molecules-28-00043-f003:**
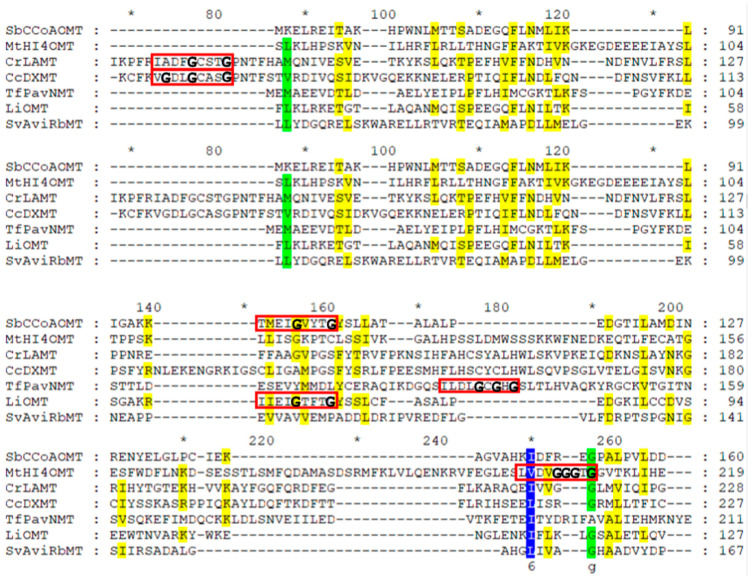
Alignment of the protein sequences of plant NPMTs and bacterial MTs. The plant species and accession numbers of these MTs are listed in Table 1. Plant NPMTs include cation-dependent SbCCoAOMT, cation-independent MtHI4′OMT, SABATH CrLAMT, and CcDXMT, and BIA TfPavNMT, in comparison to the bacterial MT within Class I bacterial LiOMT (*Leptospira interrogans*, WP_000087781), and non-Class I bacterial SvAviRbMT (*Streptomyces viridochromogenes*, WP_003998250). The red boxes indicate the glycine-rich Rossmann fold consensus sequences with 0–3 mismatches. The amino acid residues highlighted in blue are highly conserved, those in green are conserved, and those in yellow are moderately conserved. “*” represents every 10th amino acid after each numerical indicator.

**Figure 4 molecules-28-00043-f004:**
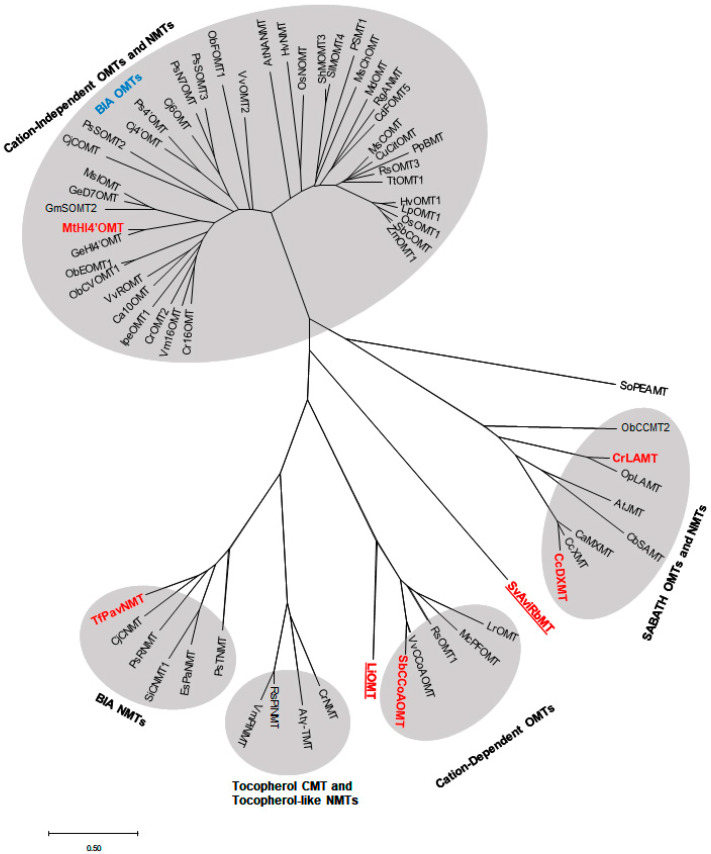
Phylogenetic relationships among functionally characterized plant NPMTs and bacterial MTs. A phylogenetic tree was generated using the neighbor-joining method and the Poisson correlation method via MEGA X [161]. The scale bar shows the amino acids substituted per sequence alignment. Bolded sequences indicate the NPMTs further analyzed in this review. The underlined sequences are bacterial methyltransferases, showing the relationship between bacteria and plant NPMTs. The plant species and accession numbers can be found in Table 1. The bacterial MT species and accession numbers are: LiOMT, *Leptospira interrogans* O-methyltransferase (WP_000087781); SvAviRbMT, *Streptomyces viridochromogenes* antibiotic-resistant mediating O-methyltransferase (WP_003998250).

**Figure 5 molecules-28-00043-f005:**
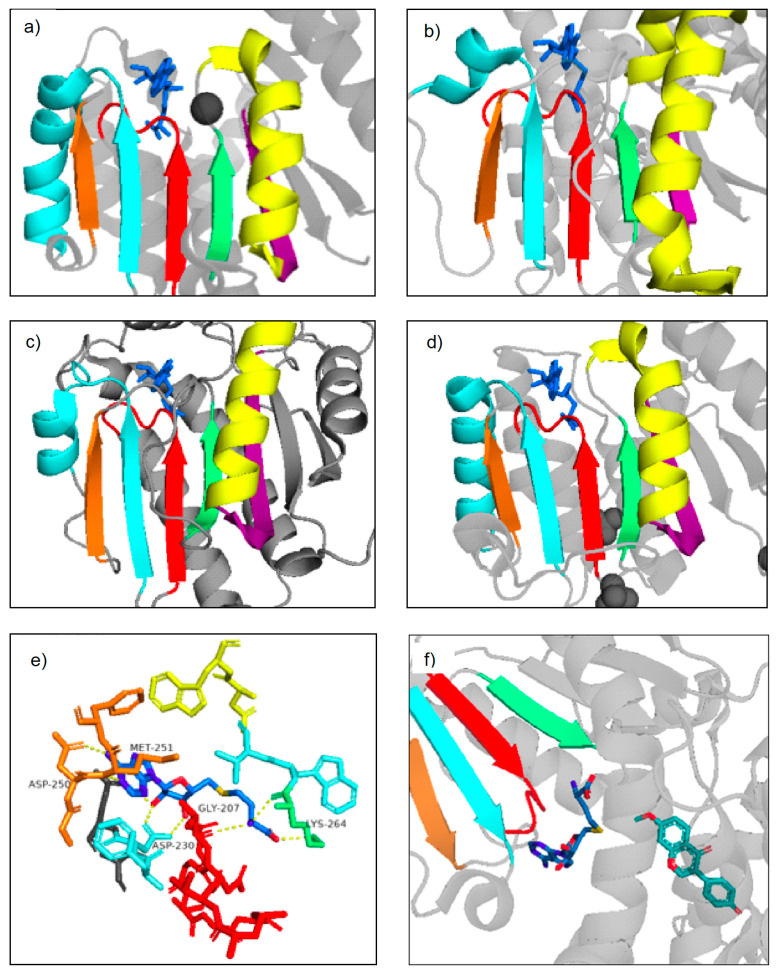
Rossmann fold motifs highlighted in Class I plant NPMTs and Class I bacterial MTs. Three-dimensional visualizations were created using PDB crystal structures and PyMOL. (**a**) Cation-dependent, SbCCoAOMT, (**b**) Cation-independent MtHI4′OMT, (**c**) SABATH, CrLAMT, (**d**) bacterial *Leptospira interrogans* O-methyltransferase, LiOMT (PDB 2HNK), (**e**) closer look at the MtHI4′OMT SAM-binding pocket, with important residues and hydrogen bonding highlighted, (**f**) representation of the methyl transfer between SAM and MtH14′OMT. Species names and PDB numbers for the above-mentioned plant NPMTs can be found in Table 1. The colored beta sheets and alpha helices represent Motifs I-VI, which make up the characteristic MT Class I Rossmann fold. The red strand represents Motif I, the light blue strand and helix represents Motif II, the orange strand represents Motif III, the green strand represents Motif IV, the yellow helix represents Motif V, and the purple strand represents Motif VI. The darker blue structure is either SAM or its demethylated form, SAH. The lighter blue structure is the substrate.

**Table 1 molecules-28-00043-t001:** Biochemically characterized plant natural product methyltransferases.

Name	Plant Species	Acceptor	PNPMT Class	Pathway/Substrate Class	Accepted Substrate	Nucleotide Accession Number	Protein Accession Number	PDB	Ref
**ALKALOIDS**									
Cj4’OMT	*Coptis japonica*	O	OMT	benzylisoquinoline	(R,S)-laudanosoline, (R,S)-6-O-methylnorlaudanosoline, (R,S)-norlaudanosoline, (S)-scoulerine	D29812	BAB08005		[34]
Cj6OMT	*Coptis japonica*	O	OMT	benzylisoquinoline	(R,S)-norococlaurine, (R,S)-6-O-methylnorlaudanosoline, (R,S)-laudanosoline, (R,S)-norlaudanosoline, laudanosoline, (S)-scoulerine	D29811	BAB08004		[34]
CjCNMT	*Coptis japonica*	N	NMT	benzylisoquinoline	(R,S)-coclaurine, (R,S)-norreticuline, (R,S)-norlaudanosoline, (R,S)-6-O-methylnorlaudanosoline, 6,7-dimethoxyl-1,2,3,4-tetrahydroisoquinoline, 1-methyl-6,7-dihydroxy-1,2,3,4-tetrahydroisoquinolinne	AB061863	BAB71802	6GKZ	[35,36]
PSMT1	*P. somniferum*	O	OMT	benzylisoquinoline	scoulerine	JQ658999	AFB74611	6I5Z	[37]
PsN7OMT	*P. somniferum*	O	OMT	benzylisoquinoline	(S)-norreticuline	FJ156103	ACN88562		[38]
PsSOMT1-3	*P. somniferum*	O	OMT	benzylisoquinoline	(S)-scoulerine, (S)-tetrahydrocolumbamine, (S)-norreticuline, (S)-reticuline	JN185323 (1) JN185324 (2) JN185325 (3)	AFK73709 (1) AFK73710 (2) AFK73711 (3)		[39]
SiSOMT	*Stephania intermedia*	O	OMT	benzylisoquinoline	(S)-scoulerine, (S)-tetrahydropalmatrubine, (S)-tetrahydrocolumbamine	MK749415	QFU85196		[40]
SiCNMT1-3	*Stephania intermedia*	N	NMT	benzylisoquinoline	(R)-coclaurine	MK749412 MK749413 MK749414	QFU85193 QFU85194 QFU85195		[40]
St6OMT1	*Stephania tetrandra*	O	OMT	benzylisoquinoline	(S)-norcoclaurine				[41]
NnOMT1,5	*Nelumbo nucifera*	O	OMT	benzylisoquinoline	1-benzylisoquinolines	XM_010245752 XM_010249599 XM_010249600 XM_010273389 XM_010277761	XP_010244054 XP_010247901 XP_010247902 XP_010271691 XP_010276063		[42]
PsRNMT	*Papaver somniferum*	N	NMT	benzylisoquinoline	(R)-reticuline, (S)-reticuline, papaverine, (R,S)-tetrahydropapaverine, boldine, (S)-corytuberine, (+)-isothebaine, (+)-isocorydine, (+)-glaucine, (+)-bulbocapnine, narcotine hemiacetal, noscapine, hydrastine	KX369612	AOR51552		[43]
PsTNMT	*P. somniferum*	N	NMT	benzylisoquinoline	(R,S)-canadine, (R,S)-tetrahydropalmatine, (R,S)-stylopine	DQ028579	AAY79177		[44]
TfPavNMT	*Thalictrum flavum*	N	NMT	benzylisoquinoline	(S)-reticuline, pavine, (R,S)-tetrahydropapaverine, (R,S)-scoulerine, (R,S)-stylopine	EU883010	ACO90251	5KOK	[45]
GfTNMT	*Glaucium flavum*	N	NMT	benzylisoquinoline	(S)-stylopine, tetrahydropalmatine, (S)-canadine, (S)-tetrahydrocolumbamine, (S)-scoulerine			6P3O	[46]
TtOMTI (Thatu OMT II;1.1)	*Thalictrum tuberosum*	O	OMT	benzylisoquinoline	caffeic acid, catechol, guajacol, ferulic acid, sinapic acid, (S)-norcoclaurine, (S)-norlaudanosoline, (R,S)-3ʹ-O-methylnorlaudanosoline, (R,S)-4ʹ-O-methylnorlaudanosoline, (R,S)-laudanosoline, (S)-4ʹ-O-methyllaudanosoline, (S)-nororientaline, (R)-nororientaline, (R,S)-norisoorientaline, (S)-reticuline, (R,S)-3-O-demethylcheilanthifoline, (S)-6-O-demethylautumnaline, (R)-6-O-demethylautumnaline	AF064693	AAD29841		[47]
Ca10OMT	*Camptotheca acuminata*	O	OMT	monoterpene indole, flavonoid, phenolics	10-hydroxycamptothecin, kaempferol, quercetin, kaempferol 3-OGlc, quercetin 3-O-Glc, 7-O-methylquercetin, 4ʹ-O-methylquercetin	MG996006	AWH62806		[48]
Cr16OMT	*Catharanthus roseus*	O	OMT	monoterpene indole	16-hydroxytabersonine	EF444544	ABR20103		[27]
CrNMT	*Catharanthus roseus*	N	NMT	monoterpene indole	16-methoxy- 2,3-dihydro-3-hydroxy tabersonine	HM584929.1	ADP00410.1		[49]
TiN10OMT	*Tabernanthe iboga*	O	OMT	monoterpene indole	ibogamine, noribogaine, 10-hydroxycoronaridine	MH454075	AXF35975 (partial)		[50]
VmPiNMT	*Vinca minor*	O	OMT	monoterpene indole	picrinine, 21-hydroxylochnericine, norajmaline	KC708450	AHH02782		[51]
RsANMT	*Rauvolfia serpentina*	N	NMT	monoterpene indole	ajmaline, norajmaline	KC708445	AHH02777		[52]
RsNNMT	*Rauvolfia serpentina*	N	NMT	monoterpene indole	norajmaline, Nb-methylnorajamaline	KC708449	AHH02781		[52]
RsPiNMT	*Rauvolfia serpentina*	N	NMT	monoterpene indole	picrinine, 21-hydroxylochnericine, norajmaline	KC708448	AHH02780		[52]
Vm16OMT	*Vinca minor*	O	OMT	monoterpene indole	16-hydroxytabersonine	MH010798	QBY35563		[53]
PiIpeOMT1-3	*Psychotria ipecacuanha*	O	OMT	monoterpene isoquinoline	isococlaurine, N-deacetylisoipecoside, 7-O-methyl-N-deacetylisoipecoside, cephaeline, norcoclaurine, 4-O-methyllaudanosoline, nororientaline, isoorientaline, (1R) norprotosinomenine, (1S) norprotosinomenine, protosinomenine	AB527082 (1) AB527083 (2) AB527084 (3)	BAI79243 (1) BAI79244 (2) BAI79245 (3)		[28]
CaDXMT1	*Coffea arabica*	N	NMT	purine	paraxanthine, theobromine, 7-methylxanthine	AB084125	BAC75663		[54]
CaMXMT	*C. arabica*	N	NMT	purine	7-methylxanthine, paraxanthine	AB048794	BAB39216		[55]
CaXMT1	*C. arabica*	N	NMT	purine	xanthine	AB048793	BAB39215		[54]
CcDXMT	*Coffea canephora*	N	SABATH	purine	3,7-dimethylxanthine	DQ422955	ABD90686	2EFJ	[56]
CcXMT	*Coffea canephora*	N	SABATH	purine	xanthosine	DQ422954	ABD90685	2EG5	[56]
CmXRS1	*C. arabica*	N	NMT	purine	xanthosine	AB034699	BAC43755		[57,58]
PcCS	*Paullinia cupana var. sorbilis*	N	NMT	purine	theobromine, 7-methylxanthine	BK008796	DAA64605		[59]
NpN4OMT1	*Narcissus* sp.	O	OMT	phenethylamine	norbelladine, *N*-methylnorbelladine, dopamine	KJ584561	AIL54541		[29]
LrOMT	*Lycoris radiata*	O	Cation-dependent OMT	alkaloid	norbelladine, caffeic acid, 3,4-dihyroxybenzaldehyde, dopamine, 3,4-dihydroxybenzylamine, higenamine, 1,2,3,4-4H-6,7-isoquinolinediol, (-)-epinephrine, (-)-norepinephrine, 5-hydroxyvanillin, 3,4,5-trihydroxybenzaldehyde, ethyl 3,4-dihydroxybenzoate, 4-Br-catechol, 4-F-catechol	MK805029	QEP29044		[60]
PMT	*Nicotiana tabacum*	N	NMT	amine	putrescine	D28506	BAA05867		[61]
PMT	*Solanum tuberosum*	N	NMT	amine	putrescine	AJ605553	CAE53633		[62]
PMT	*Calystegia sepium*	N	NMT	amine	putrescine	AM177608	CAJ46252		[63]
PMT	*Datura innoxia*	N	NMT	amine	putrescine	AM177609 AM177610	CAJ46253 CAJ46254		[63]
PMT	*Physalis divaricata*	N	NMT	amine	putrescine	AM177611	CAJ46255		[63]
PMT	*Datura stramonium*	N	NMT	amine	putrescine	AJ583514	CAE47481		[64]
**PHENOLICS**									
CdFOMT5	*Citrus depressa*	O	OMT	flavonoid	quercetin, 3-hydroxyflavone, 5-hydroxyflavone, 6-hydroxyflavone, 7-hydroxyflavone, naringenin, (-)-epicatechin, equol	LC126059	BAU51794		[65]
CrOMT2	*Catharanthus roseus*	O	OMT	flavonoid	myricetin, quercetin, dihydroquercetin, dihydromyricetin	AY127568	AAM97497		[66]
CrOMT6	*Catharanthus roseus*	O	OMT	flavonoid	homoeriodictyol, isorhamnetin, chrysoeriol, quercetin, eriodictyol, kaempferol	AY343490	AAR02420		[67]
CuCitOMT	*Citrus unshiu Marc.*	O	OMT	flavonoid	3′,4′-dihydroxyflavone, 3′,4′,5,7-tetrahydroxyflavone	LC516612	BBU25484		[68]
HvOMT1	*Hordeum vulgare*	O	OMT	flavonoid	tricetin, luteolin, tricetin, quercetin, 5-hydroxyferulic acid, eriodictyol, taxifolin	EF586876	ABQ58825		[69]
ZmOMT1	*Zea mays*	O	OMT	flavonoid	luteolin, tricetin, quercetin, 5-hydroxyferulic acid, eriodictyol, taxifolin	XM_002436508	ABQ58826		[69]
ObF8OMT-1	*Ocimum basilicum*	O	OMT	flavonoid	7,8,4ʹ-OH-flavone, 8-OH-7-OCH_3_-flavone, 7,8-OH-flavone, 7,8,3ʹ,4ʹ-OH-flavone	KC354402	AGQ21572		[70]
ObFOMT1-6	*Ocimum basilicum*	O	OMT	flavonoid	luteolin, apigenin, scutellarein, hispidulin, naringenin, chrysoeriol, diosmetin, acacetin, scutellarein-4ʹ-methyl ether, nevadensin, cirsimaritin, kaempferol, quercetin, scutellarein-4ʹ-methyl ether, nepetin, ladanein, cirsioliol, genkwanin, scutellarein-7-methyl ether, naringenin-7-methyl ether, scutellarein-7-O-glucuronide	JQ653275 (1) JQ653276 (2) JQ653277 (3) JQ653278 (4) JQ653279 (5) JQ653280 (6)	AFU50295 (1) AFU50296 (2) AFU50297 (3) AFU50298 (4) AFU50299 (5) AFU50300 (6)		[70]
ObPFOMT-1	*Ocimum basilicum*	O	OMT	flavonoid	7,8,4ʹ-OH-flavone, 7,8-OH-flavone, 6,7-OH-flavone, 5,6-OH-flavone, 5,6-OH-7-OCH_3_-flavone, eriodictyol, ladanein, scutellarein-7-methyl ether, scutellarein 4ʹ-methyl ether, scutellarein, scutellarin, cirsiliol, nepetin, luteolin, luteolin-7-methyl ether, luteoline-7-glucoside, quercetagetin, quercetin, quercetin-7-methyl ether, tricetin, 3ʹ,4ʹ-OH-flavone, 5,3ʹ,4ʹ-OH-flavone, 7,3ʹ,4ʹ-OH-flavone, 7,8,3ʹ,4ʹ-OH-flavone	KC354401	AGQ21571		[70]
OsNOMT	*Oryza sativa*	O	OMT	flavonoid	racemic naringenin, kaempferol, apigenin, luteolin, racemic liquiritigenin, quercetin	AB692949	BAM13734		[71]
ShMOMT3	*Solanum habrochaites*	O	OMT	flavonoid	quercetin, kaempferol, myricetin, 7-methyl quercetin, 3-methyl quercetin, 3-methyl myricetin, 3ʹ,5ʹ-dimethyl myricetin, 3ʹ-methyl quercetin	KC513419	AGK26768		[72]
SlMOMT4	*Solanum lycopersicum*	O	OMT	flavonoid	myricetin, 3′-methylmyricetin	KF740343	AIN36846		[73]
MpOMT4	*Mentha* x *piperita*	O	OMT	flavonoid	isorhamnetin, kaempferol, quercetin, rhamnetin, luteolin, apigenin, 6-OH-apigenin, 7,8,3ʹ,4ʹ-OH-flavone, naringenin, taxifolin	AY337461	AAR09602		[74]
PaF4’OMT	*Plagiochasma appendiculatum*	O	OMT	flavonoid	apigenin, luteoline, scutellarein, genkwanin, eriodictyol, naringenin, quercetin, kaempferol, genistein	KY977687	ARS23163		[75]
ShMOMT1	*Solanum habrochaites*	O	OMT	flavonoid	myricetin, quercetin, 7-methyl quercetin, 3-methyl quercetin	JF499656	ADZ76433		[76]
ShMOMT2	*Solanum habrochaites*	O	OMT	flavonoid	7-methyl quercetin, quercetin, kaempferol, myricetin, 4ʹ-methyl kaempferol, 3,7,4ʹ-trimethyl kaempferol, 3ʹ-methyl quercetin, 3-methyl quercetin, 3,7,3ʹ,4ʹ-tetramethyl quercetin, 3ʹ-methyl myricetin, 3ʹ,5ʹ-dimethyl myricetin, 3ʹ,4ʹ,5ʹ-trimethyl myricetin	JF499657	ADZ76434		[76]
AtCCoAOMT7	*Arabidopsis thaliana*	O	OMT	flavonoid	luteolin, quercetin, caffeoyl-CoA, esculetin	At4g26220	NP_567739		[77]
AtOMT1	*Arabidopsis thaliana*	O	OMT	flavonoid	quercetin, myricetin, luteolin	U70424	AAB96879		[78,79]
CaFOMT1	*Chrysosplenium americanum*	O	OMT	flavonoid	3,7,4ʹ-triOMeQ, 2ʹ-OH 3,6,7,4ʹ-tetraOMeQg, 2ʹ-OH 3,7,4ʹ-triOMeQ, 3,6,7,4ʹ-tetraOMeQg; Abbreviations: Q, quercetin; Qg, quercetagetin (6-OH-Q)	U16794	AAA80579		[80]
CaOMT2	*Chrysosplenium americanum*	O	OMT	flavonoid	quercetin, luteolin, 5-hydroxyferulic, caffeic acid	U16793	AAA86982		[81]
OsCAldOMT1	*Oryza sativa*	O	OMT	flavonoid	5-hydroxyconiferaldehyde, selgin		Q6ZD89		[82]
VvOMT1-2	*Vitis vinifera*	O	OMT	flavonoid	quercetin, resveratrol, caffeic acid, epicatechin, 3-isobutyl-2-methoxypyrazine, 3-isopropyl-2-methoxypyrazine	GQ357167 (1) GQ357168 (2)	ADJ66850 (1) ADJ66851 (2)		[31]
EnFOMT	*Eucalyptus nitida*	O	OMT	flavonoid	pinocembrin, chrysin, naringenin, apigenin, alpinetin, 7-hydroxyflavone, hesperetin, luteolin, quercetin	OM96491	UOO01100		[83]
GmSOMT-2	*Glycine max*	O	OMT	flavonoid	naringenin, daidzein, quercetin, genistein, apigenin	TC178411 (TIGR)			[84]
VvAOMT2	*Vitis vinifera*	O	Cation-dependent OMT	anthocyanin	delphinidin 3-*O*-glucoside, cyanidin 3-*O*-glucoside	HQ702997	ADY18303		[85]
VvCCoAOMT	*Vitis vinifera*	O	Cation-dependent OMT	anthocyanin	cyanidin 3-O-glucoside chloride, caffeoyl-CoA	Z54233	CAA90969		[86,87,88]
GeD7OMT	*Glycyrrhiza echinata*	O	OMT	isoflavone	daidzein	AB091685	BAC58012		[89]
GeHI4ʹOMT	*Glycyrrhiza echinata*	O	OMT	isoflavone	(2R,3S)-2,7,4ʹ-trihydroxyisoflavanone, medicarpin	AB091684	BAC58011		[89]
GmIOMT1	*Glycine max*	O	OMT	isoflavone	6-hydroxydaidzein, 8-hydroxydaidzein, 3ʹ-hydroxydaidzein	NM_001250549	NP_001237478		[90]
MsIOMT	*Medicago sativa*	O	OMT	isoflavone	6,7,4ʹ-trihydroxyisoflavone, daidzein, genistein, (+)6a-hydroxymaackiain, (+)maackiain	AF000976	AAC49927	1FP2	[91,92]
MsI7OMT	*Medicago truncutula*	O	OMT	isoflavone	6,7,4ʹ-trihydroxyisoflavone, daidzein, (+)6a-hydroxymaackiain			6CIG	[92]
MtHI4ʹOMT	*Medicago truncutula*	O	OMT	isoflavone	(2S, 3R)-2,7,4ʹ-trihydroxyisoflavanone, 6a-hydroxymaackiain	AY942158	AAY18581	1ZG3 1ZHF	[8]
PmIOMT9	*Pueraria montana* var. lobata	O	OMT	isoflavone	genistein, daidzen, prunetin, isoformononetin	KP057892	AKW47171		[93]
PIOMT4	*Pueraria lobata*	O	OMT	isoflavone	3ʹ-hydroxy-daidzein, luteolin, quercetin	KP057887	AKW47166		[94]
McPFOMT	*Mesembryanthemum crystallinum*	O	Cation-dependent OMT	phenylpropanoid	quarcetagetin, quercetin, caffeoyl coA, caffeic acid	AY145521	AAN61072	3C3Y	[95]
GmSOMT-9	*Glycine max*	O	Cation-dependent OMT	phenylpropanoid	quercetin, luteolin, taxifolin, catechin, taxifolin, caffeic acid	NM_001249311	NP_001236240		[96]
MsCOMT	*Medicago sativa*	O	OMT	phenylpropanoid	5-hydroxyconiferaldehyde, caffeic acid, 5-hydroxyferulic acid, caffeoyl aldehyde, caffeoyl alcohol, 5-hydroxyconiferyl alcohol	M63853	AAB46623	1KYW 1KYZ	[97]
ObCVOMT1	*Ocimum basilicum*	O	OMT	phenylpropanoid	chavicol, eugenol, t-ioseugenol, t-anol, catechol, phenol, coniferyl alcohol	AF435007	AAL30423		[98]
ObEOMT1	*O. basilicum*	O	OMT	phenylpropanoid	eugenol, chavicol, t-ioseugenol, guaiacol, caffeic acid, coniferyl alcohol, ferulic acid	AF435008	AAL30424		[98]
OsROMT9	*Oryza sativa*	O	OMT	phenylpropanoid	quercetin, catechin, eriodictyol, luteolin, myricetin, taxifolin, rhamnetin, caffeic acid	DQ288259	ABB90678		[99]
OsOMT1	*Oryza sativa*	O	OMT	phenylpropanoid	tricetin, luteolin, quercetin, eriodictyol, 5-hydroxyferulic acid	DQ530257	ABF72191		[100]
VpOMT4	*Vanilla planifolia*	O	Cation-dependent OMT	phenylpropanoid	caffeoyl coA	JF344740	ADZ76153		[101]
RsOMT1,3	*Rauvolfia serpentina*	O	Cation-dependent OMT	phenylpropanoid	caffeic acid, 3,5-dimethoxy-4-hydroxycinnamic, 3,4,5-trihydroxybenzoic	KX687823 (1) KX687825 (3)	AOZ21151 (1) AOZ21153 (3)		[102]
MsCCoAOMT	*Medicago sativa*	O	Cation-dependent OMT	phenylpropanoid	caffeoyl CoA	U20736	AAC28973	1SUI	[103]
SbCCoAOMT	*Sorghum bicolor*	O	Cation-dependent OMT	phenylpropanoid	caffeoyl-CoA	XM_002436505	XP_002436550	5KVA	[104]
SbCOMT	*Sorghum bicolor*	O	OMT	phenylpropanoid	5-hydroxyconiferaldehyde, caffeic acid, p-coumaraldehyde, coniferaldehyde	XM_002436506	ADW65743		[105]
VvOMT3	*Vitis vinifera*	O	OMT	phenylpropanoid	3-isopropyl-2-hydoxypyrazine, 3-isobutyl-2-hydroxypryazine, quercetin, resveratrol, caffeic acid, epicatechin, catechin, eugenol, isoeugenol, orcinol	XM_002436507	AGK93042		[32]
ObCCMT1-3	*Ocimum basilicum*	O	SABATH	phenylpropanoid	*trans*-cinnamic acid, hydrocinnamic acid, *p*-coumaric, 4-hydroxyhydrocinnamic acid, *m*-courmaric acid, benzoic acid	XM_002436509	ABV91100 (1) ABV91101 (2) ABV91102 (3)		[106]
LpCaOMT	*Lolium perenne*	O	Cation-dependent OMT	phenylpropanoid	caffeoyl alcohol, caffeic acid, 5-hydroxyferulic acid, caffeoyl aldehyde, 5-hydroxyconiferaldehyde	AF033538	AAD10253	3P9C 3P9I 3P9K	[107]
MOMT4	*Clarkia breweri*	O	OMT	phenylpropanoid	coniferyl alcohol, sinapyl alcohol	JX287369	AFQ94040	3TKY	[108]
HlOMT1-2	*Humulus lupulus*	O	OMT	chalcone	desmethylxanthohumol, xanthohumol	EU309725 (1) EU309726 (2)	ABZ89565 (1) ABZ89566 (2)		[109]
MtChOMT	*Medicago truncutula*	O	OMT	chalcone	2′,4,4′-trihydroxychalcone	L10211	AAB48059	1FPQ	[92,110]
AcOMT1	*Acorus calamus*	O	OMT	polyphenol	isorhapontigenin, resveratrol, piceatannol, oxyresveratrol, pinostilbene, naringenin, anol, isoeugenol, chavicol, eugenol, p-coumaric acid, caffeic acid	LC387636	BBE32341		[111]
VvROMT	*Vitis vinifera*	O	OMT	polyphenol	resveratrol monomethyl ether, resveratrol	FM178870	CAQ76879		[112]
AtSAMT	*Arabidopsis thaliana, Arabidopsis lyrata*	O	SABATH	phenolics	benzoic acid, salicylic acid, anthranilic acid	AY224595 AY224596	AAP57210 AAP57211		[113]
CbSAMT	*Clarkia breweri*	O	SABATH	phenolics	salicylic acid, benzoic acid	AF133053	AAF00108	1M6E	[5,23]
**MONOTERPENES**									
CrLAMT	*Catharanthus roseus*	O	SABATH	monoterpene iridoid	loganic acid, secologanic acid	EU057974	ABW38009	6C8R	[24]
OpLAMT	*Ophiorrhiza pumila*	O	SABATH	monoterpene iridoid	loganic acid, secologanic acid	MT942677	QWX38535		[114]
**FURANOCOUMARIN**									
PpBMT	*Peucedanum praeruptorum*	O	OMT	furanocoumarin	bergaptol	KU359196	ANA75355	5XG6 5XOH	[115]
**TOCOPHEROLS**									
AtγTMT	*Arabidopsis thaliana*	C	CMT	vitamin E	δ-tocopherol, γ-tocopherol	AF104220	AAD02882		[116,117]
Pfγ-TMT	*Perilla frutescens*	C	CMT	vitamin E	γ-tocopherol	AF213481	AAL36933		[118]
**POLYKETIDE**									
MdOMT	*Malus domestica*	O	OMT	polyketide	3,5-dihydroxybiphenyl	MF740747	ASV64939		[119]
**STEROLS**									
GmSMT	*Glycine max*	C	CMT	sterol	sterol	U43683	AAB04057		[120]
TwSMT1	*Tripterygium wilfodii*	C	CMT	sterol	sterol	KU885950	ARI48333		[121]
Ntsmt2-1	*Nicotiana tabacum*	C	CMT	sterol	24-methylene lophenol	U71108 U81312	AAB62808 AAC34951		[122,123]
Ntsmt1-1	*Nicotiana tabacum*	C	CMT	sterol	cycloartenol	AF053766	AAC35787		[123]
**HALIDE/THIOCYANATE**									
AtHOL1	*Arabidopsis thaliana*	SCN, halides	HTMT	thiocyanate/halide	SCN- > I > Br > Cl (Not F)	AY044314	AAK73255	3LCC	[22,124]
RsHTMT	*Raphanus sativus*	Halides	HTMT	thiocyanate/halide	halides (except F)	AB477013	BAH84870		[125]
**AMINOBENZOATE**									
RgANMT	*Ruta graveolens*	N	NMT	aminobenzoate	anthranilate	DQ884932	ABI93949		[126]
**OTHER SMALL MOLECULES**									
AtASMT	*Arabidopsis thaliana*	O	OMT	indole	*N*-acetylserotonin, serotonin	AT4G35160	Q9T003		[127]
EsPaNMT	*Ephedra sinica*	N	NMT	monoamine alkaloid	(+/-)-cathinone, (+/-)-norephedrine, (-)-norpseudoephedrine, (+/-)-ephedrine, (+)-pseudoephedrine	MH029305	AWJ64115		[128]
HvNMT	*Hordeum vulgare*	N	NMT	indole	3-aminomethylindole, 3-aminomethylindole, N-methyl-3-aminomethylindole	U54767	AAC18643		[129]
SoPEAMT	*Spinacia oleracea*	N	NMT	phospholipid	phosphatidylethanolamine	AF237633	AAF61950		[130]
DTCMT	*Brassica rapa*	S	SMT	organosulfur	dithiocarbamate	Brara.B01660 Brara.G00303 (Phytozome)	Brara.B01660 Brara.G00303 (Phytozome)		[131]
AtJMT	*Arabidopsis thaliana*	O	SABATH	carboxylic acid	(±) jasmonic acid, dihydrojasmonic acid	AY008434	AAG23343		[7]
CrSMT1	*Catharanthus roseus*	S	SMT	organosulfur	benzene thiol, furfuryl thiol, 3-mercaptohexyl-acetate, 3-mercaptohexan-1-ol, benzoyl thiol, 1-mercaptopropan-2-ol, pyridine-2-thiol, phenol, 1,3-hexandiol, 1,4-dithiothreitol, 3-mercaptopropan-1-ol, 6-mercapto-hexan-1-ol, 2-mercaptoethanol (BME)	DQ084384	AAZ32409		[132]
EjMBMT	*Eriobotyra japonica*	O	SABATH	carboxylic acid	*p*-methoxybenzoic acid, benzoic acid, jasmonic acid	LC127197	BAV54103		[133]
AtIAMT	*Arabidopsis thaliana*	O	SABATH	indole	indole acetic acid	NM_124907	NP_200336	3B5I	[134]
BoSMT	*Brassica oleracea*	Se	SeMT	amino acid	*DL*-selenocysteine, *L*-selenocysteine, *DL*-cysteine, *L*-cysteine, *DL*-homocysteine	AY817737	AAX20123		[135]

## Data Availability

Not applicable.

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
