# Peer review of "Functional Diversification and Structural Origins of Plant Natural Product Methyltransferases"

_molecules, 2022, doi:10.3390/molecules28010043_

Round 1

Reviewer 1 Report

This paper is a comprehensive review of the known biochemically characterized plant natural product methyltransferases (PNPMTs) regarding their functional and structural diversity. It summarises PNPMTs regarding their classes and substrate specificity with diverse product structures, plant species, DNA, protein and PDB given, which is a valuable summary and reference for readers. It discusses the amino acid sequence features of PNPMTs and classifies them based on phylogenetic analysis; Structural features of PNPMTs and how they interact with the cofactor SAM and different substrates are being discussed; And examples of protein engineering to alter regioselectivity and substrate specificity of PNPMTs are given. The review paper covers important topics in this field and I’d recommend accepting the paper in the present form.

Author Response

Thank you very much for your comments.

Reviewer 2 Report

The paper is well written and structured, but there are still some aspects that should be revised.

- Introduction section -  must be completed with a detailed description of the work's novelty.

- Figure 1 -  is not clear, it must be redone.

- Figure 2 -  is not clear, it must be redone.

- Conclusions and Future Directions section - must be completed and structured, so as to present the aspects researched so far and the directions that are recommended for the future.

Author Response

Thank you for your comments and suggestions. We have made the following modifications:

1. Introduction section.

We have added more description suggested in the introduction section.

2. Figure 1 and 2 Clarity

Figure 1 and 2 – we have improved the spacing between individual molecules for enhanced visualization.  We highlighted the target atoms instead of the methyl group for clarity. We also modified the figure titles for accuracy.

Figure 1. Representation of chemical diversity generated by O-directed methyltransferases. Methylated atoms are highlighted in yellow.

Figure 2. Representation of chemical diversity generated by N-, C-, S-directed methyltransferases. Methylated atoms are highlighted in blue (N-), green (C-), and red (S-).

3. Conclusions and Future Directions section 

We have added more description suggested in the conclusions and future directions section.
